# Retrospective Analysis of Factors Associated with the Treatment Outcomes of Intradiscal Platelet-Rich Plasma-Releasate Injection Therapy for Patients with Discogenic Low Back Pain

**DOI:** 10.3390/medicina59040640

**Published:** 2023-03-23

**Authors:** Koji Akeda, Tatsuhiko Fujiwara, Norihiko Takegami, Junichi Yamada, Akihiro Sudo

**Affiliations:** Department of Orthopaedic Surgery, Mie University Graduate School of Medicine, Tsu 514-8507, Japan

**Keywords:** intervertebral disc degeneration, low back pain, platelet-rich plasma, intradiscal injection therapy

## Abstract

*Background and Objectives*: Recently, the clinical application of platelet-rich plasma (PRP) has gained popularity for the treatment of degenerative disc diseases. However, the regenerative effects and factors associated with treatment outcomes after intradiscal injection of PRP remain unknown. This study aimed to evaluate time-dependent changes in imaging findings related to intervertebral disc (IVD) degeneration and to identify factors associated with the outcomes of PRP injection therapy. *Materials and Methods*: A retrospective analysis of a previous randomized clinical trial of intradiscal injection of the releasate isolated from PRP (PRPr) in patients with discogenic low back pain (LBP) was performed. Radiographic parameters (segmental angulation and lumbar lordosis) and MRI phenotypes, including Modic changes, disc bulge, and high-intensity zones (HIZs), were evaluated at baseline and 6 and 12 months post-injection. Treatment outcomes were evaluated based on the degree of LBP and LBP-related disability at 12 months post-injection. *Results*: A total of 15 patients (mean age: 33.9 ± 9.5 years) were included in this study. Radiographic parameters showed no significant changes after the PRPr injection. There were no remarkable changes in the prevalence or type of MRI phenotype. Treatment outcomes were significantly improved after treatment; however, the number of targeted discs and the presence of posterior HIZs at baseline were significantly but negatively associated with treatment outcomes. *Conclusions*: Intradiscal injection of PRPr significantly improved LBP and LBP-related disability 12 months post-injection; however, patients with multiple target lesions or posterior HIZs at baseline were significantly associated with poor treatment outcomes.

## 1. Introduction

Low back pain (LBP) is the leading cause of years lived with disability in 126 countries and the leading cause of worldwide productivity loss [1]. The annual expenditure for the management of LBP is estimated to exceed USD 100 billion in the United States of America [2].

Among the anatomical elements of the lumbar spine, intervertebral disc (IVD) degeneration is a major contributing factor in LBP [3]. The IVD lies between adjacent vertebrae interposed with a cartilaginous endplate and consists of an outer annulus fibrosus (AF) and an inner nucleus pulposus (NP). The outer AF consists of collagen-rich fibrocartilage lamellae, while the inner NP is composed of a proteoglycan-rich gel-like matrix, and both provide structural stability and exhibit shock absorption properties [3].

IVD degeneration is characterized by both cellular and extracellular matrix changes within the degenerating IVD, which progressively cause structural failure and impaired mechanical and physiological IVD functions [4]. Enhanced expression of pro-inflammatory cytokines (Interleukin-1β [IL-1β] and tumor necrosis factor-α [TNF-α]) found in degenerated human IVDs induces progressive degradation of major extracellular matrix components by stimulating matrix-degrading enzymes [4] that induce disc ruptures, including several types of AF tears [5,6,7]. Progression of IVD degeneration gives rise to intervertebral instability and eventually affects lumbar spinal alignment. The endplate and vertebral bone marrow adjacent to degenerated IVDs also represent tissue changes in response to IVD degeneration [8,9]. These changes, along with IVD degeneration, are believed to be associated with LBP [3,4,10].

The extent of IVD degeneration is clinically evaluated by magnetic resonance imaging (MRI) [11,12]. Lumbar MRI phenotypes, including Modic changes [9,13], disc bulges [14,15], and high-intensity zones (HIZs) [16,17,18], are often interpreted as causes of LBP. Radiographic parameters, including intervertebral instability assessed by segmental angulation [19] and lumbar sagittal imbalance assessed by lumbar lordosis (LL) [20], have also been reported to be associated with LBP.

Basic research on IVD regeneration has been conducted globally, and considerable progress has been made in clinical applications using biological approaches [21,22,23] and cell therapy [24]. Platelet-rich plasma (PRP) is an autologous blood concentrate containing mostly bioactive molecules [25], which has been used for the treatment of musculoskeletal disorders to enhance tissue regeneration and repair [22]. It has been reported that PRP has the potential to stimulate the metabolism of IVD cells in vitro and regenerative effects on degenerated IVDs in several animal models [22]. Recently, the clinical application of PRP has gained popularity in the treatment of degenerative disc diseases. A recent meta-analysis of clinical studies showed a significant improvement in LBP following PRP injection [26].

A previous randomized, double-blind, active-controlled clinical trial evaluated the efficacy and safety of the releasate isolated from PRP (PRPr) injection into degenerated discs of patients with discogenic LBP [27]. The intradiscal injection of PRPr showed clinically significant improvements in LBP intensity, similar to those injected with corticosteroids (CS) at 8 weeks post-injection. It was concluded that PRPr treatment was safe and maintained improvements in pain and LBP-related disability for 60 weeks. However, whether intradiscal administration of PRPr induces tissue repair and/or regenerative effects, which improve the image findings associated with disc degeneration, remains unknown. Furthermore, whether the radiographic and/or MRI findings associated with disc degeneration significantly affect the outcomes of intradiscal injection of PRPr has not been evaluated.

The aims of this study were as follows: (1) to retrospectively evaluate time-dependent changes in radiographic parameters (segmental angulation and lumbar lordosis) and MRI phenotypes (Modic changes, disc bulge, and HIZs) after intradiscal injection of PRPr for the patients with discogenic LBP; and (2) to identify the factors associated with the outcomes of PRPr treatment.

## 2. Materials and Methods

### 2.1. Study Design and Patients

This study was a retrospective analysis of a previous randomized, double-blind, active-controlled clinical trial conducted between February 2018 and September 2020 [27]. This retrospective study was approved by the Clinical Research Ethics Review Committee of the Mie University Hospital (Approval number: H2023-001). Informed consent was determined by way of an opt-out form on the website.

Patients received an intradiscal injection of either PRPr or corticosteroids (CS). Patients from both the PRPr and CS groups who still experienced pain received PRPr as optional treatment at 8 weeks post-injection. Among the 16 patients, 15 received optional injections of PRPr 8 weeks after the initial injection. As this study aimed to investigate the factors associated with the outcomes of PRPr treatment, one patient in the CS group who received only one injection of CS was excluded from the current retrospective analysis (Figure 1). The efficacy of PRPr was evaluated for up to 12 months (52 weeks) after optional injection. Patients aged > 18 years who had LBP for more than 3 months with 1 or more lumbar discs (L3/L4 to L5/S1) with evidence of degeneration, as indicated by MRI, and at least 1 symptomatic disc, confirmed using standardized provocative discography, were included in the study. The inclusion and exclusion criteria have been previously reported [27].

### 2.2. PRP Releasate Preparation and Fluoroscopy-Guided Injection

PRPr was isolated as previously reported [28]. Briefly, autologous PRP was prepared from whole blood (400 mL) using the buffy coat method [29]. The supernatant isolated from activated PRP (PRPr) was stored at −20 °C until use.

Intravenous antibiotics (cefazolin sodium, 1 g) were administered within 60 min before the injection procedure. The injection site was treated with local anesthetic (0.5% lidocaine). Under fluoroscopy, a 22-gauge, 150 mm spinal needle was inserted into the center of the targeted disc. PRPr (2 mL) or CS (betamethasone sodium phosphate, Sionogi & Co., Ltd., Osaka, Japan) (2 mg in 2.0 mL of saline) was injected through a syringe filter.

### 2.3. Outcome Measures

The visual analog scale (VAS) (0–100 mm) [30,31] at baseline and 12 months after the optional injection (Post 12 M) was measured to evaluate the extent of LBP. The Oswestry Disability Index (ODI) [32] and Roland–Morris Disability Questionnaire (RDQ) [33] were measured at baseline and 12 months after the optional injection (Post 12 M) to evaluate LBP-related disability. The change (Baseline − Post 12 M) and % change ([baseline − Post 12 M]/baseline × 100) in VAS, ODI, and RDQ were also evaluated.

### 2.4. Radiographic Evaluation

Radiographic data at baseline, Post 6 M and Post 12 M were retrospectively evaluated by 1 orthopedic surgeon (N.T.) who was blinded to the treatment group. To evaluate intervertebral instability, segmental angulation was measured as the difference in intervertebral angle from extension to flexion [34]. Lumbar lordosis (LL) angles [35] were evaluated using the Cobb angle between the L1 superior endplate and S1 superior endplate for the evaluation of the sagittal alignment of the lumbar spine.

### 2.5. MR Imaging

Lumbar MRI was performed using a 1.5 T MR scanner (Ingenia, Philips, Best, The Netherlands) with the patient in the supine position. The imaging protocol included sagittal T2W turbo-spine echo (TSE) with a repetition time (TR) of 2500 ms/echo and an echo time (TE) of 85 ms. The field of view (FOV) was 300 × 300 mm. All sagittal slice intervals were 4.4 mm thick, and a total of 17 slices were available.

### 2.6. MRI Evaluation

MRI data at baseline and Post 12 M were retrospectively evaluated by an orthopedic surgeon (T.F.) who was blinded to the treatment group. The extent of IVD degeneration of the targeted discs at baseline and Post 12 M was evaluated using a modified Pfirrmann [11] grading system. Modic and vertebral body marrow changes adjacent to the endplate were evaluated as previously reported [9,13,36]. Type I was defined by areas of hypointensity on T1-weighted (T1W) and hyperintensity on T2-weighted (T2W) images, type II by areas of hyperintensity on both T1W and T2W images, and type III by areas of hypointensity on both T1W and T2W images. Disc bulge is a generalized extension of the disc beyond the edges of the vertebral ring apophyses [37]. In this study, disc bulge was diagnosed as discs extending into the spinal canal beyond the edge of the apophyses at the mid-sagittal section of T2W images of the lumbar spine. An HIZ was diagnosed as a high-intensity signal identified within the AF with a low-intensity signal on T2W MR images of the lumbar spine [18]. The presence of HIZs was evaluated on consecutive sagittal MR images (two or more adjacent sagittal images) at both the anterior and posterior AF [38]. The morphology of HIZs was classified into five types (round, fissure, vertical, rim, and enlarged), as previously reported [39].

### 2.7. Statistical Analysis

A power analysis revealed that a sample size of 9 patients per treatment group was necessary to detect a substantial difference between the PRPr and CS groups in the VAS at week 8 with 80% power and a two-sided significance of 0.05. Differences in the basic characteristics between the treatment groups were analyzed using an unpaired *t*-test or chi-square test. Differences in VAS, ODI, and RDQ scores at baseline and Post 12 M among the groups were analyzed by unpaired *t*-test or 1-way analysis of variance (ANOVA). Correlations between radiographic parameters and VAS, ODI, and RDQ scores and changes in these parameters and % change were analyzed using a Pearson correlation coefficient test. Time-dependent changes in segmental angulation and LL angle were assessed for statistical significance using two-way repeated-measures ANOVA. Multiple regression analyses were performed to identify factors contributing to VAS at Post 12 M, change in VAS, % change in VAS, ODI at Post 12 M, change in ODI, % change in ODI, RDQ at Post 12 M, change in RDQ, and % change in RDQ as a dependent variable with image characteristics at baseline (number of targeted discs, MRI grade, Modic change, disc bulge, anterior HIZ, posterior HIZ, segmental angulation, and LL angle), and treatment group. Data are expressed as mean ± standard deviation (SD). All statistical analyses were performed using IBM Statistical Package for Social Sciences Software (SPSS) Statistics version 28.0 (IBM Japan, Tokyo, Japan). The accepted level of significance was set at *p* < 0.05.

## 3. Results

### 3.1. Patient Characteristics

A total of 15 patients (mean age: 33.9 ± 9.5 years, 11 men, 4 women) were included in this study. The CS group included 6 patients (mean age: 32.2 ± 6.8 years), and the PRPr group included 9 patients (mean age: 34.9 ± 11.3 years). A total of 19 discs were used for the treatment. The patient characteristics are summarized in Table 1. No significant differences in age, sex, number of targeted discs, modified Pfirrmann classification, intervertebral angle, or LL angle were found between the groups. There were no significant differences in the prevalence of Modic changes, disc bulge, or posterior HIZs by patient or disc level between the groups. The prevalence of anterior HIZs by disc level, but not by patient, was significantly higher in the CS group than in the PRPr group (*p* < 0.05) (Table 1).

### 3.2. Change in Segmental Angulation of Targeted Discs

At baseline, the mean segmental angulation was 14.7 ± 2.7 in the CS group and 14.8 ± 6.8 in the PRPr group (*p* = 0.83; Table 1). There were no significant changes in the mean segmental angulation of each group during the observational period (6 M: CS: 13.9 ± 4.6, PRPr: 18.3 ± 7.6; 12 M: CS: 14.0 ± 5.4, PRPr: 13.5 ± 6.2, *p* = 0.63). No significant differences were found between the groups (*p* = 0.33). There was no significant interaction between the groups and time points (*p* = 0.36).

### 3.3. Change in Lumbar Lordosis Angle

At baseline, the mean LL angle was 49.1 ± 8.1 in the CS group and 40.4 ± 14.1 in the PRPr group, and this difference was not significant (*p* = 0.15; Table 1). The LL angles of each group did not show any significant changes throughout the observational period (6 M: CS: 45.5 ± 5.6, PRPr: 38.5 ± 14.1; 12 M: CS: 49.4 ± 7.5, PRPr: 42.3 ± 13.3, *p* = 0.47) and did not differ significantly between the groups (*p* = 0.22). There was no significant interaction between groups and time points (*p* = 0.87).

### 3.4. Modic Changes

Modic changes adjacent to the targeted discs were found in 21.1% of the targeted discs (4 discs in 3 patients) at baseline (2 discs in the CS group and 2 discs in the PRPr group) (Table 1). Among them, Modic types 2 and 3 were found on the 2 discs. There were no significant differences in the prevalence of Modic changes between the CS and PRPr groups at baseline (Table 1). The prevalence and types of Modic changes did not change at Post 6 M and 12 M (baseline: type 2 [*n* = 2], type 3 [*n* = 2]; 6 M: type 2 [*n* = 2], type 3 [*n* = 2]; 12 M: type 2 [*n* = 1], type 3 [*n* = 2], 1 patient dropped out of the study).

### 3.5. Disc Bulge

A disc bulge was found in 63.2% of the targeted discs (12 discs in 11 patients) at baseline. There was no significant difference in the prevalence of disc bulge between the PRPr and CS groups at baseline (*p* = 0.53, Table 1). The prevalence of disc bulge did not change at Post 6 M and 12 M (6 M: 12 discs [63.2%]; 12 M: 10 discs [58.8%], 1 dropped out of the study).

### 3.6. High-Intensity Zones in Anterior AF

HIZs in the anterior AF (anterior HIZs) were identified in 26.3% of the targeted discs (5 discs in 4 patients) at baseline. No significant difference in prevalence was observed between the CS and PRPr groups (Table 1). Among them, the round type was found in four discs, and the rim type in one disc. The anterior HIZs at baseline did not change during follow-up; however, 1 round type and 1 rim type were newly identified at Post 12 M.

### 3.7. High-Intensity Zones in Posterior AF

HIZs in the posterior AF (posterior HIZs) were identified in 68.4% of the targeted discs (13 discs in 11 patients) at baseline. No significant difference in prevalence was observed between the CS and PRPr groups (Table 1). A total of 3 discs (15.8%) were of the round type, and 10 discs (52.6%) were of the vertical type. After shrinking, 1 vertical-type HIZ in the posterior AF disappeared at Post 12 M. Other HIZs showed no changes in prevalence or type.

### 3.8. Factors Associated with VAS

There were no significant differences in VAS at baseline among the groups classified according to the image characteristics of the targeted discs, including the number of targeted discs (a), modified Pfirrmann grading (b), Modic changes (c), disc bulge (d), anterior (e) and posterior (f) HIZs, and treatment group (g) (Figure 2a–g). However, the VAS at Post 12 M was significantly higher in patients without Modic changes compared to those with Modic changes (*p* < 0.05, Figure 2c) and in patients with posterior HIZs compared to those without posterior HIZs (*p* < 0.05, Figure 2f).

The change in VAS score was significantly lower in patients with posterior HIZs than in those without posterior HIZs (*p* < 0.05, Table 2). The % change in VAS was significantly lower in patients with 2 targeted discs compared to those without 2 targeted discs, significantly lower in patients without Modic changes compared to those with Modic changes, and significantly lower in patients with posterior HIZs compared to those without posterior HIZs (all *p* < 0.05; Table 2). Segmental angulation or LL angle at baseline showed no significant correlations with VAS at Post 12 M, change in VAS, or % change in VAS.

### 3.9. Factors Associated with ODI

No significant differences in the ODI at baseline were found among the groups classified according to the image characteristics of the targeted discs at baseline and treatment group (Figure 3a–g). However, the ODI at Post 12 M was significantly higher in patients without Modic changes compared to those with Modic changes (*p* < 0.05, Figure 3c) and patients with posterior HIZs compared to those without posterior HIZs (*p* = 0.05, Figure 3f).

At baseline, there were no significant differences in the change in ODI between the groups classified according to the image characteristics of the targeted discs or treatment group (Table 3). The % change in ODI was significantly lower in patients without Modic changes compared to those with Modic changes (*p* < 0.05) and in patients with posterior HIZs compared to those without posterior HIZs (*p* < 0.05; Table 3). Segmental angulation or LL angle at baseline showed no significant correlations with ODI at Post 12 M, change in ODI, or % change in ODI.

### 3.10. Factors Associated with RDQ

At baseline, no significant differences in RDQ scores were found among the groups classified according to the baseline characteristics of the targeted discs or according to the treatment group (Figure 4). However, the RDQ at Post 12 M in the CS group was significantly higher than that in the PRPr group (*p* < 0.05, Figure 4g).

There were no significant differences in the change in RDQ scores among the groups classified according to the baseline characteristics of the targeted discs (Table 4). The % change in RDQ was significantly lower in patients without Modic changes compared to those with Modic changes (*p* < 0.05) and in patients with posterior HIZs compared to those without posterior HIZs (*p* < 0.05) (Table 4). Segmental angulation or LL angle at baseline showed no significant correlations with RDQ at Post 12 M, change in RDQ, or % change in RDQ.

### 3.11. Multiple Regression Analysis for Identifying the Factors Associated with VAS, ODI and RDQ

Multiple regression analysis for identifying the factors associated with VAS revealed that the number of targeted discs was significantly associated with the % change in VAS score (*p* < 0.01, Table 5c). Regarding the ODI, the analysis identified that the presence of posterior HIZs was a significant factor associated with ODI at Post 12 M (*p* = 0.05, Table 5d). ‘Posterior HIZs’ and a ‘number of targeted discs’ were the factors significantly associated with the % change in ODI (both *p* < 0.05, Table 5f). Regarding the RDQ, the analysis identified that the ‘treatment group’ was a significant factor associated with the RDQ at Post 12 M (*p* < 0.05, Table 5g). ‘Treatment group’ was also significantly associated with a % change in RDQ (*p* < 0.05, Table 5i).

## 4. Discussion

In this study, among the 15 patients, 11 (73.3%) had 1 targeted disc, and 4 (26.7%) had 2 targeted discs. Univariate analysis showed that patients treated with the two targeted discs showed a significantly lower % change in VAS. Multivariate analysis showed that ‘two targeted discs’ was significantly associated with a decrease in % change in VAS and ODI, suggesting that the number of targeted discs may affect the outcomes of PRPr treatment.

The targeted discs were classified from grades 4 to 6 using the Modified Pfirrmann grading system [11], which reflects moderate IVD degeneration. The results of this study showed that the MRI grades limited from grades 4 to 6 had no significant effects on the outcome of PRPr injection therapy for discogenic LBP; however, the differences in the treatment outcome may be found in the case of advanced stages of degeneration.

LL is the anterior curvature of the lumbar spine formed by the wedging of lumbar IVDs, and its angle is an important parameter of lumbar sagittal balance [35]. A meta-analysis by Chun et al. [20] demonstrated that LBP was strongly associated with a decreased LL curve. Another factor considered to be related to LBP is intervertebral instability [19]. This current study evaluated the LL angle and intervertebral instability using lateral lumbar radiographs and showed that neither parameter showed any significant changes after PRPr injection. Furthermore, neither parameter affected the treatment outcomes of PRPr injections. Although the mean age of the patients in our study was relatively young, these parameters may affect the treatment outcomes in middle-aged and older patients with advanced stages of IVD degeneration.

A previous population-based study showed that Modic changes in the lumbar region were found in 63.5% of the middle-aged population [8]. Among them, the prevalence of Modic type II was the highest, followed by types I and II. Importantly, only Modic type I changes were significantly associated with LBP. In our study, Modic type II or III was found only in 20% of patients, whose VAS and ODI scores were significantly lower compared to those without Modic changes at Post 12 M. As the prevalence of Modic changes was small in our study, this result should be considered as a reference. However, it is considered that discogenic patients with Modic type II or III changes can be included as candidates for PRPr treatment.

Disc bulges are often identified on lumbar MRI of asymptomatic populations [15]; however, several studies have demonstrated that disc bulges are significantly associated with LBP [14,40]. Disc bulge was identified in 73% of patients with discogenic LBP; however, it was not significantly related to the extent of LBP or LBP-related quality of life at baseline. In addition, the presence of disc bulges did not affect either the change or % change in VAS, ODI, or RDQ. The authors speculated that disc bulges in the targeted discs would have little impact on the treatment outcomes of PRPr injection.

Originally, HIZs were defined as high-intensity signals on T2W MRI of the posterior AF [18]. HIZs are considered important biomarkers of discogenic LBP because they correlate with induced pain after provocative discography [18]. However, there is controversy regarding the clinical significance of HIZs because some studies have demonstrated that HIZs are also frequently found in asymptomatic subjects [41]. A recent large population-based cohort study showed that lumbar HIZs, especially homogenous multilevel HIZs, are significantly associated with prolonged and/or severe LBP and sciatica [42]. This current study identified posterior HIZs in 73% of the patients with discogenic LBP. The results of this current study demonstrated that the presence of posterior HIZs had no significant effect on the extent of LBP and LBP-related disability at baseline; however, VAS and ODI scores at Post 12 M were significantly higher in patients with posterior HIZs than in those without them. Multiple regression analysis revealed that posterior HIZs were significant factors associated with ODI at Post 12 M and % change in ODI. These results suggest that posterior HIZs do not affect LBP intensity itself but may affect the treatment outcomes of IVD reparative therapies, including PRPr injection.

Recently, a significant association between anterior HIZs and discogenic LBP has also been reported [43]; however, anterior HIZs were found only in four patients in our study. Moreover, the presence of anterior HIZs did not affect the extent of LBP and LBP-related disabilities either at baseline or Post 12 M, suggesting that anterior HIZs may not affect the treatment outcomes of PRPr injection therapy.

A previous randomized controlled study reported that the % change in RDQ score in PRPr patients was significantly higher than that in CS patients at week 32 post-optional injection [27]. Consequently, the subjects in the PRPr group received two injections of PRPr, whereas the subjects in the CS group received a single injection of corticosteroid and PRPr. In this current study, multiple regression analysis revealed that the ‘treatment group’ was significantly associated with RDQ at Post 12 M and % change in RDQ. The results suggest that LBP-related disability was significantly improved by the two injections of PRPr compared to the injection of CS followed by PRPr. Therefore, two injections of PRPr are recommended for patients with discogenic LBP where further improvements in LBP are expected.

In this current study, LBP-related disability was assessed using both the ODI and RDQ. Both assessment scores were improved by PRPr treatment at Post 12 M; however, a statistical significance was identified only in the RDQ score. Chiarotto et al. [44] conducted a meta-analysis to compare the measurement properties of the ODI and RDQ in adult patients with non-specific LBP. They reported that the ODI showed better reliability and less measurement errors, whereas the RDQ showed better construct validity as a measure of physical functioning. The authors speculated that the improvement of physical functioning in daily life activities was better reflected by RDQ than ODI after the injection of PRPr.

This study had several limitations. First, because this study aimed to investigate the factors associated with PRPr treatment, one patient in the CS group who received only CS was excluded. Second, patients in the CS group underwent CS followed by PRPr. It is unclear whether CS influences the subsequent administration of PRPr. Therefore, it was not possible to compare the treatment outcomes of single or double injections of PRPr in this study. Thirdly, there was an imbalance in the number of male and female patients in this study. Therefore, the possibility that gender differences affect the efficacy of PRPr treatment cannot be denied. Finally, the number of subjects in this study was small, and the number of subjects in the two groups was uneven. Therefore, the results obtained in this study should be regarded as preliminary data.

## 5. Conclusions

The results of this current study showed that radiographic parameters and/or MRI phenotypes associated with IVD degeneration did not show remarkable changes 12 months after PRPr injection into painful (targeted) IVDs. This current study identified that the number of painful discs (targeted discs) was one of the factors associated with poor treatment outcomes. Therefore, the number of painful discs (targeted discs) by patients should be considered to accurately evaluate treatment in future clinical studies of disc therapies. This current study also noted that the presence of posterior HIZs was associated with poor LBP-related disability. Elucidating the pathology of posterior HIZs may lead to further improvements in the treatment of discogenic LBP.

## Figures and Tables

**Figure 1 medicina-59-00640-f001:**
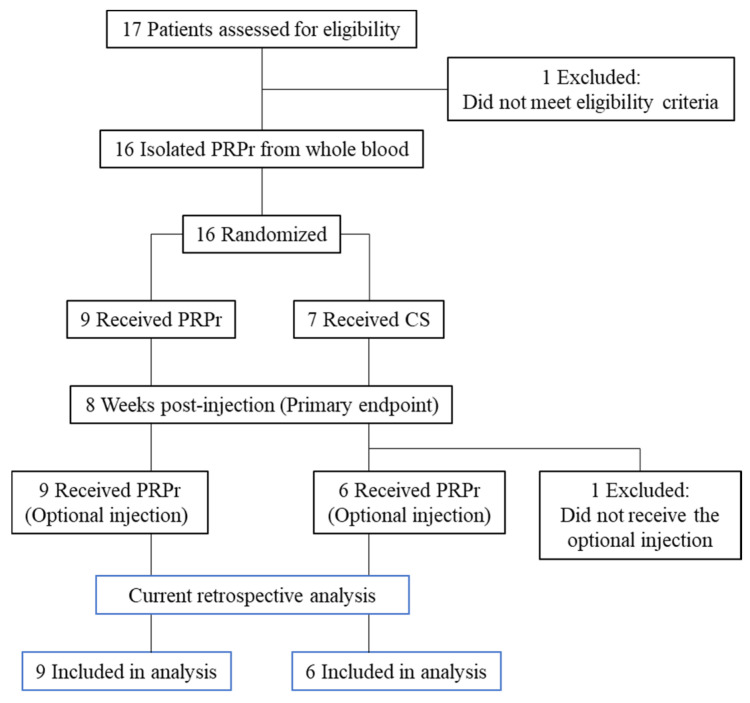
Flowchart of patients. Patients received an intradiscal injection of either the platelet-rich plasma releasate (PRPr) or corticosteroids (CS).

**Figure 2 medicina-59-00640-f002:**
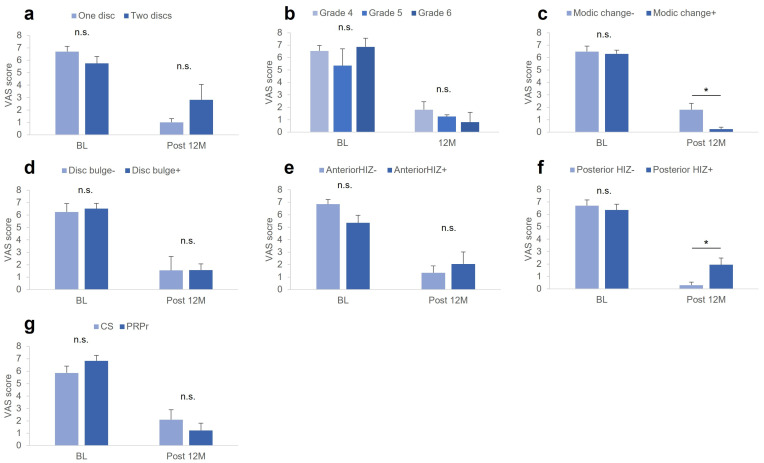
VAS at baseline and Post 12 M. VAS scores at baseline and Post 12 M among the groups classified by image characteristics of the targeted discs, including (**a**) number of targeted discs, (**b**) modified Pfirrmann grading, (**c**) Modic changes, (**d**) disc bulge, (**e**) anterior HIZ, (**f**) posterior HIZs, and (**g**) treatment group. VAS, visual analog scale; CS, corticosteroid; PRPr, platelet-rich plasma releasate; HIZ, high-intensity zone. * *p* < 0.05 between the groups.

**Figure 3 medicina-59-00640-f003:**
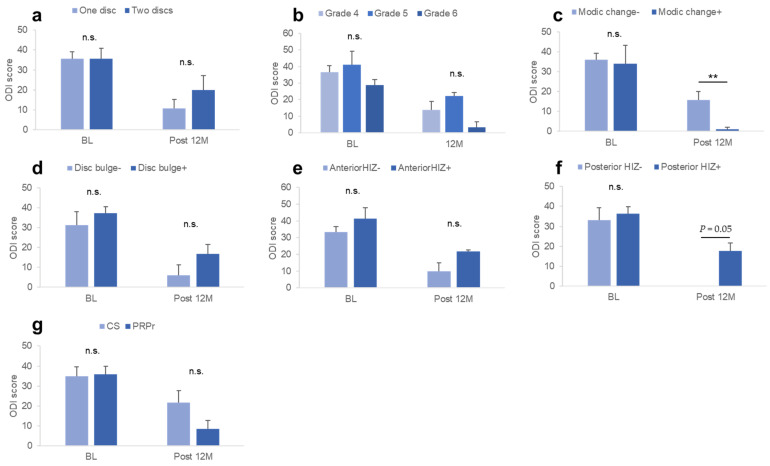
ODI at baseline and Post 12 M. ODI at baseline and Post 12 M among the groups classified by the image characteristics of the targeted discs, including (**a**) number of targeted discs, (**b**) modified Pfirrmann grading, (**c**) Modic changes, (**d**) disc bulge, (**e**) anterior HIZ, (**f**) posterior HIZs, and (**g**) treatment group. ODI, Oswestry Disability Index; CS, corticosteroid; PRPr, platelet-rich plasma releasate; HIZ, high-intensity zone. ** *p* < 0.05 between the groups.

**Figure 4 medicina-59-00640-f004:**
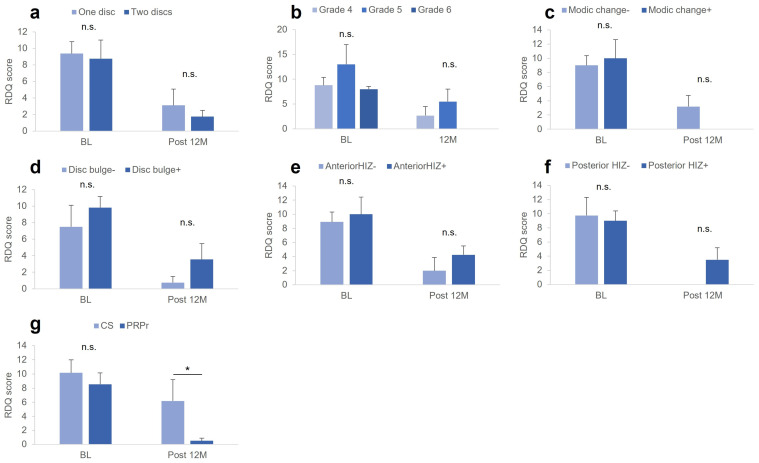
RDQ at baseline and Post 12 M. RDQ scores at baseline and Post 12 M among the groups classified by image characteristics of the targeted discs, including (**a**) number of targeted discs, (**b**) modified Pfirrmann grading, (**c**) Modic changes, (**d**) disc bulge, (**e**) anterior HIZ, (**f**) posterior HIZs, and (**g**) treatment group. CS, corticosteroid; RDQ, Roland–Morris Disability Questionnaire; PRPr, platelet-rich plasma releasate; HIZ, high-intensity zone. * *p* < 0.05 between the groups.

**Table 1 medicina-59-00640-t001:** Patient’s baseline characteristics.

	Total	CS	PRPr	*p*-Value
Number of patients (discs)	15 (19)	6 (8)	9 (11)	
Age	33.9 ± 9.5	32.2 ± 6.8	34.9 ± 11.3	0.59
Sex (Male/Female)	11/4	5/1	6/3	0.46
Number of the target disc				0.63
One disc	11	4	7	
Two discs	4	2	2	
Modified Pfirrmann classification				0.18
Grade 4	10	3	7	
Grade 5	2	2	0	
Grade 6	3	1	2	
Segmental angulation	14.7 ± 5.4	14.7 ± 2.7	14.8 ± 6.8	0.99
Lumbar lordosis angle	43.9 ± 12.5	49.1 ± 8.1	40.4 ± 14.1	0.15
Modic change				
By patient	3/15 (20%)	2/6 (33.3%)	1/9 (11.1%)	0.34
By disc level	4/19 (21.1%)	2/8 (25%)	2/11 (18.2%)	0.57
Disc Bulge				
By patient	11/15 (73.3%)	4/6 (66.7%)	7/9 (77.8%)	0.54
By disc	12/19 (63.2%)	5/8 (62.5%)	7/11 (63.6%)	0.66
Anterior HIZ				
By patient	4/15 (26.7%)	3/6 (50%)	1/9 (11.1%)	0.14
By disc	5/19 (26.3%)	4/8 (50%)	1/11 (9.1%)	<0.05
Posterior HIZ				
By patient	11/15 (73.3%)	5/6 (83.3%)	6/9 (66.7%)	0.46
By disc	13/19 (68.4%)	7/8 (87.5%)	6/11 (54.5%)	0.18

CS: corticosteroid, PRPr: platelet-rich plasma releasate, HIZ: high-intensity zone.

**Table 2 medicina-59-00640-t002:** Factors associated with VAS.

Associated Factor	Change in VAS				*p*-Value	% Change in VAS				*p*-Value
**Number of the Target Disc**	One		Two			One		Two		
	5.1 (2.3)		2.9 (3.5)		0.31	84.1 (12.8)		43.3 (53.1)		<0.05
**MRI grade**	G4	G5		G6		G4	G5		G6	
	4.5 (2.7)	2.6 (3.9)		6.3 (2.8)	0.46	67.3 (40.8)	75.8 (4.7)		86.4 (19.2)	0.8
**Modic change**	-		+			-		+		
	4.2 (3.0)		6.0 (0.9)		0.16	67.1 (6.0)		95.7 (3.9)		<0.05
**Disc bulge**	-		+			-		+		
	4.7 (3.4)		4.3 (2.7)		0.86	69.2 (49.2		72.6 (29.8		0.91
**Anterior HIZ**	-		+			-		+		
	5.3 (2.5)		2.5 (3.1)		0.18	77.3 (31.7)		58.5 (42.3)		0.46
**Posterior HIZ**	-		+			-		+		
	6.4 (3.8)		3.8 (2.9)		<0.05	96.1 (5.5)		64.2 (36.5)		<0.05
**Group**	CS		PRPr			CS		PRPr		
	3.0 (2.9)		5.3 (2.5)		0.18	61.0 (37.9)		78.2 (33.2)		0.42

Change (Baseline − Post 12 M) and % change ([baseline − Post 12 M]/baseline × 100) in visual analog scale (VAS) of each associated factor are shown. MRI grade was evaluated by modified Pfirrmann [11] grading system. HIZ: high-intensity zone. Number in parentheses indicates standard deviation (SD).

**Table 3 medicina-59-00640-t003:** Factors associated with ODI.

Associated Factor	Change in ODI				*p*-Value	% Change in ODI				*p*-Value
**Number of the Target Disc**	One		Two			One		Two		
	24.4 (7.5)		15.6 (24.0)		0.32	75.9 (27.7)		34.3 (46.1)		0.17
**MRI grade**	G4	G5		G6		G4	G5		G6	
	21.7 (16.5)	19.0 (14.2)		24.3 (3.0)	0.94	61.6 (42.3)	43.0 (23.0)		90.0 (14.1)	0.5
**Modic change**	-		+			-		+		
	19.4 (12.1)		34.5 (23.3)		0.52	57.3 (38.6)		95.0 (7.1)		<0.05
**Disc bulge**	-		+			-		+		
	25.1 (19.5)		20.2 (12.3)		0.66	76.9 (39.7)		57.0 (37.9)		0.43
**Anterior HIZ**	-		+			-		+		
	15.1 (13.0)		18.0 (13.0)		0.54	73.8 (40.0)		39.0 (19.9)		0.06
**Posterior HIZ**	-		+			-		+		
	33.9 (14.8)		18.1 (12.4)		0.2	100 (0)		52.0 (36.6)		0.05
**Group**	CS		PRPr			CS		PRPr		
	13.9 (9.7)		26.5 (14.8)		0.09	42.4 (31.5)		76.0 (37.6)		0.11

Change (baseline − Post 12 M) and % change ([baseline − Post 12 M]/baseline × 100) in Oswestry Disability Index (ODI) [32] of each associated factor are shown. MRI grade was evaluated by modified Pfirrmann [11] grading system. HIZ: high-intensity zone. Number in parentheses indicates standard deviation (SD).

**Table 4 medicina-59-00640-t004:** Factors associated with RDQ.

Associated Factor	Change in RDQ			*p*-Value	% Change in RDQ			*p*-Value
**Number of the Target Disc**	One	Two			One	Two		
	7.0 (5.4)	7.0 (7.5)		1.0	77.1 (41.3)	74.2 (21.1)		0.9
**MRI grade**	G4	G5	G6		G4	G5	G6	
	6.8 (6.3)	7.5 (2.1)	7.5 (0.7)	0.98	74.6 (40.7)	59.8 (9.7)	100 (0)	0.55
**Modic change**	-		+		-	+		
	6.4 (5.0)		10.5 (6.4)	0.52	71.9 (37.0)	100 (0)		<0.05
**Disc bulge**	-		+		-	+		
	6.8 (5.7)		7.1 (5.3)	0.92	87.5 (25.0)	71.2 (39.4)		0.39
**Anterior HIZ**	-		+		-	+		
	7.6 (6.1)		5.8 (2.5)	0.46	84.3 (40.2)	58.0 (7.9)		0.09
**Posterior HIZ**	-		+		-	+		
	10.0 (6.2		6.1 (4.8)	0.4	100 (0)	36.1 (37.7)		<0.05
**Group**	CS		PRPr		CS	PRPr		
	4.2 (4.5		8.8 (5.0)	0.13	49.6 (44.4)	92.8 (14.1)		0.1

Change (baseline − Post 12 M) and % change ([baseline − Post 12 M]/baseline × 100) in Roland–Morris Disability Questionnaire (RDQ) [33] of each associated factor are shown. MRI grade was evaluated by modified Pfirrmann [11] grading system. HIZ: high-intensity zone. Number in parentheses indicates standard deviation (SD).

**Table 5 medicina-59-00640-t005:** Results of multiple regression analyses.

a. VAS at 12 Months Post-injection (VAS Post 12 M)				
n.s.				
b. Change in VAS at 12 months Post-injection (Change in VAS)				
n.s.				
c. Percent Change in VAS at 12 months Post-injection (% change in VAS)				
	β	βstand	T	*p* value
Number of targeted discs	−40.7	−0.57	−2.3	<0.05
R^2^: 0.32, *p* < 0.05				
d. ODI at 12 months Post-injection (ODI Post 12 M)				
	β	βstand	T	*p* value
Posterior HIZ	17.6	0.55	2.2	0.05
R^2^: 0.31, *p* = 0.05				
e. Change in ODI at 12 months Post-injection (Change in ODI)				
n.s.				
f. Percent Vhange in ODI at 12 Months Post-injection (% Change in ODI)				
	β	βstand	T	*p* value
Posterior HIZ	−49.4	−0.57	−2.9	<0.05
Number of targeted discs	−43	−0.54	−2.7	<0.05
R^2^: 0.60, *p* = 0.010.				
g. RDQ at 12 Months Post-injection (RDQ Post 12 M)				
	β	βstand	T	*p* value
Treatment group	−5.7	−0.59	−2.4	<0.05
R^2^: 0.35, *p* < 0.05				
h. Change in RDQ at 12 Months Post-injection (Change in RDQ)				
n.s.				
i. Percent Change in RDQ at 12 months Post-injection (% Change in RDQ)				
	β	βstand	T	*p* value
Treatment group	43.2	0.62	2.6	<0.05
R^2^: 0.38, *p* < 0.05				

VAS: visual analogue scale; ODI: Oswestry Disability Index [32]; RDQ: Roland–Morris Disability Questionnaire [33].

## Data Availability

The data presented in this study are available on reasonable request from the corresponding author.

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
