# Peer review of "Retrospective Analysis of Factors Associated with the Treatment Outcomes of Intradiscal Platelet-Rich Plasma-Releasate Injection Therapy for Patients with Discogenic Low Back Pain"

_medicina, 2023, doi:10.3390/medicina59040640_

Round 1

Reviewer 1 Report

1.       The introduction is very well grounded, but please emphasize the elements of originality/ novelty that this study brings to the specialized literature. The aims of this study were: (1) to retrospectively evaluate time-dependent changes in radiographic parameters (segmental angulation and lumbar lordosis) and MRI phenotypes (Modic changes, disc bulge, and HIZs) after intradiscal injection of PRPr in a previous randomized clinical trial [27]; and (2) to identify the factors associated with the outcomes of PRPr treatment.

·         The purpose of the paper should not be cited, because then the paper will lose its originality.

·         Please rephrase and better highlight the purpose of this study;

·         Please insert the novelties that this study brings at the end of the introduction;

2.       This study was a retrospective analysis of a previous randomized, double-blind, active-controlled clinical trial conducted between February 2018 and September 2020 [27]. This retrospective study was approved by the Clinical Research Ethics Review Committee of the Mie University Hospital (Approval number: H2023-001). Informed consent was determined by way of an opt-out form on the website.

·         Again, the typology of the study is cited with the same source (27), although the two studies have different typologies: This study was a long-term follow-up of a previous prospective clinical trial, primarily a safety assessment, conducted between April 2009 and March 2012 [33]. This long-term follow-up study was approved by the Clinical Research Ethics Review Committee of Mie University Hospital. Informed consent was obtained in written form or in the opt-out form on the website. I understand it's all your article, but it doesn't seem right.

·         I repeat, using source number 27 so often diminishes the originality of the article.

·         Please kindly enter at least the sample size calculation, statistical power or both.

3.       The inclusion and exclusion criteria have been previously reported [27]. The inclusion and exclusion criteria are not contained in this article (27). Please specify inclusion and exclusion criteria.

4.       Reading the authors' article, the one in source number 27, I find that they did the same there, citing a study of their own, which again makes the originality of this article questionable: Akeda, K.; Ohishi, K. ; Masuda, K.; Bae, W.C.; Takegami, N.; Yamada, J.; Nakamura, T.; Sakakibara, T.; Kasai, Y.; Sudo, A. Intradiscal Injection of Autologous Platelet-Rich Plasma Releasate to Treat Discogenic Low Back Pain: A Preliminary Clinical Trial. Asian Spine J. 2017, 11, 380-389.

5.       After reading the article Akeda, K.; Ohishi, K. ; Masuda, K.; Bae, W.C.; Takegami, N.; Yamada, J.; Nakamura, T.; Sakakibara, T.; Kasai, Y.; Sudo, A. Intradiscal Injection of Autologous Platelet-Rich Plasma Releasate to Treat Discogenic Low Back Pain: A Preliminary Clinical Trial. Asian Spine J. 2017, 11, 380-389, I found that the two samples do not match at all, although a double reference is made there: Patients who were receiving diagnostic discography for suspected discogenic low back pain from April 2009 to March 2012 were recruited. Among 27 patients who received diagnostic discography, 14 were included in this study. Inclusion criteria for this study were being older than 18 years and having (1) chronic low back pain without leg pain for more than 3 months, (2) one or more lumbar discs (L3/L4 to L5/S1) with evidence of degenerative changes as per MRI (disc degeneration was defined as more than grade III via the Pfirrmann disc degeneration grade/classification system [15]), (3) maintenance of 50% or more of normal disc height, and (4) at least one symptomatic disc confirmed using standardized provocative discography and/or disc block. Exclusion criteria included abnormal neurological symptoms (e.g., radiculopathy) with lumbar spinal stenosis or spondylolisthesis and inflammatory arthritis (e.g., discitis).

Author Response

Response to reviewer’s comments

Reviewer 1:

  1. The introduction is very well grounded, but please emphasize the elements of originality/ novelty that this study brings to the specialized literature. The aims of this study were: (1) to retrospectively evaluate time-dependent changes in radiographic parameters (segmental angulation and lumbar lordosis) and MRI phenotypes (Modic changes, disc bulge, and HIZs) after intradiscal injection of PRPr in a previous randomized clinical trial [27]; and (2) to identify the factors associated with the outcomes of PRPr treatment.

The purpose of the paper should not be cited, because then the paper will lose its originality.

Please rephrase and better highlight the purpose of this study.

Response) Thank you very much for the reviewer’s important comments. According to the reviewer’s comment, the description was revised as follows.

Lines 79 to 83:

The aims of this study were: (1) to retrospectively evaluate time-dependent changes in radiographic parameters (segmental angulation and lumbar lordosis) and MRI phenotypes (Modic changes, disc bulge, and HIZs) after intradiscal injection of PRPr for the patients with discogenic LBP; and (2) to identify the factors associated with the outcomes of PRPr treatment.

Please insert the novelties that this study brings at the end of the introduction;

Response) According to the reviewer’s comment, the following descriptions were added to the Introduction section.

Lines 73 to 78:

However, whether intradiscal administration of PRPr induces tissue repair and/or regenerative effects, which improve the image findings associated with disc degeneration, remains unknown. Furthermore, whether the radiographic and/or MRI findings associated with disc degeneration significantly affect the outcomes of intradiscal injection of PRPr has not been evaluated.

  1. This study was a retrospective analysis of a previous randomized, double-blind, active-controlled clinical trial conducted between February 2018 and September 2020 [27]. This retrospective study was approved by the Clinical Research Ethics Review Committee of the Mie University Hospital (Approval number: H2023-001). Informed consent was determined by way of an opt-out form on the website.

Again, the typology of the study is cited with the same source (27), although the two studies have different typologies: This study was a long-term follow-up of a previous prospective clinical trial, primarily a safety assessment, conducted between April 2009 and March 2012 [33]. This long-term follow-up study was approved by the Clinical Research Ethics Review Committee of Mie University Hospital. Informed consent was obtained in written form or in the opt-out form on the website. I understand it's all your article, but it doesn't seem right.

I repeat, using source number 27 so often diminishes the originality of the article.

Response) Thank you very much for pointing out an important issue. The following description were revised as follows.

Line 91: Patients received an intradiscal injection of either PRPr or corticosteroids (CS).

Please kindly enter at least the sample size calculation, statistical power or both.

Response) According to the reviewer’s comment, the following description was added to the Statistical Analysis section.

Lines 159 to 162:

A power analysis revealed that a sample size of nine patients per treatment group was necessary to detect a substantial difference between the PRPr and CS groups in the VAS at week 8 with 80% power and a two-sided significance of 0.05.

  1. The inclusion and exclusion criteria have been previously reported [27]. The inclusion and exclusion criteria are not contained in this article (27). Please specify inclusion and exclusion criteria.

Response) The inclusion and exclusion criteria are shown in Table 1 (27). Please confirm it.

  1. Reading the authors' article, the one in source number 27, I find that they did the same there, citing a study of their own, which again makes the originality of this article questionable: Akeda, K.; Ohishi, K. ; Masuda, K.; Bae, W.C.; Takegami, N.; Yamada, J.; Nakamura, T.; Sakakibara, T.; Kasai, Y.; Sudo, A. Intradiscal Injection of Autologous Platelet-Rich Plasma Releasate to Treat Discogenic Low Back Pain: A Preliminary Clinical Trial. Asian Spine J. 2017, 11, 380-389.

Response) Thank you very much for the reviewer’s suggestion. The authors will be careful in the future.

  1. After reading the article Akeda, K.; Ohishi, K. ; Masuda, K.; Bae, W.C.; Takegami, N.; Yamada, J.; Nakamura, T.; Sakakibara, T.; Kasai, Y.; Sudo, A. Intradiscal Injection of Autologous Platelet-Rich Plasma Releasate to Treat Discogenic Low Back Pain: A Preliminary Clinical Trial. Asian Spine J. 2017, 11, 380-389, I found that the two samples do not match at all, although a double reference is made there: Patients who were receiving diagnostic discography for suspected discogenic low back pain from April 2009 to March 2012 were recruited. Among 27 patients who received diagnostic discography, 14 were included in this study. Inclusion criteria for this study were being older than 18 years and having (1) chronic low back pain without leg pain for more than 3 months, (2) one or more lumbar discs (L3/L4 to L5/S1) with evidence of degenerative changes as per MRI (disc degeneration was defined as more than grade III via the Pfirrmann disc degeneration grade/classification system [15]), (3) maintenance of 50% or more of normal disc height, and (4) at least one symptomatic disc confirmed using standardized provocative discography and/or disc block. Exclusion criteria included abnormal neurological symptoms (e.g., radiculopathy) with lumbar spinal stenosis or spondylolisthesis and inflammatory arthritis (e.g., discitis).

Response) Thank you for the reviewer’s careful suggestion. The patients of study 1 (Asian Spine J. 2017, 11, 380-389) and study 2 (J Clin Med 2022 Vol. 11 Issue 2) are not included in same cohort. Furthermore, inclusion criteria and exclusion criteria of study 1 and 2 are different. The authors appreciate your understanding.

Reviewer 2 Report

The manuscript titled, “Analysis of factors associated with the treatment outcomes of intradiscal platelet-rich plasma - releasate injection therapy for  patients with discogenic low back pain”,  retrospectively the outcomes of the previously conducted clinical trial. As stated by the authors, this study is a preliminary study, with few shortcomings, which are clearly mentioned by the authors. Even though this study can be considered preliminary, they authors have elucidated the correlations between various parameters which will be useful for clinicians and orthopedicians in deciding upon the treatment modalities and further research in the future. The manuscript can be accepted in its current form, except for very minor corrections as below.

1.      The authors could add the term “retrospective analysis” in the title of the manuscript.

2.      Line 27, ‘keyword’ itself has been mentioned as a keyword. Kindly revise.

3.      Line 43, 44, “Enhanced expression of pro-inflammatory cytokines…..”, The authors could mention the major or few of the cytokines involved in the degenerative IVD.

4.      Line 47-48, “….eventualy affects lumbar spinal malalignment.” Is it alignment or malalignement. Check and revise.

5.      Figure 1, “7 received BSP”, kindly mention the full form of BSP.

Author Response

Response to reviewer’s comments

Reviewer 2:

The manuscript titled, “Analysis of factors associated with the treatment outcomes of intradiscal platelet-rich plasma - releasate injection therapy for patients with discogenic low back pain”, retrospectively the outcomes of the previously conducted clinical trial. As stated by the authors, this study is a preliminary study, with few shortcomings, which are clearly mentioned by the authors. Even though this study can be considered preliminary, they authors have elucidated the correlations between various parameters which will be useful for clinicians and orthopedicians in deciding upon the treatment modalities and further research in the future. The manuscript can be accepted in its current form, except for very minor corrections as below.

Response) Thank you very much for the reviewer’s understanding to this manuscript.

  1. The authors could add the term “retrospective analysis” in the title of the manuscript.

Response) According to the reviewer’s comment, the tile was revised as follows.

Retrospective analysis of factors associated with the treatment outcomes of intradiscal platelet-rich plasma - releasate injection therapy for patients with discogenic low back pain

  1. Line 27, ‘keyword’ itself has been mentioned as a keyword. Kindly revise.

Response) It was corrected accordingly.

  1. Line 43, 44, “Enhanced expression of pro-inflammatory cytokines…..”, The authors could mention the major or few of the cytokines involved in the degenerative IVD.

Response) According to the reviewer’s comment, Interleukin-1β [IL-1β] and tumor necrosis factor-α [TNF-α] were added.

  1. Line 47-48, “….eventualy affects lumbar spinal malalignment.” Is it alignment or malalignement. Check and revise.

Response) Thank you for the reviewer’s suggestion. “malalignment” has change to “alignment”.

  1. Figure 1, “7 received BSP”, kindly mention the full form of BSP.

Response) BSP was changed to CS (corticosteroid).

Reviewer 3 Report

1. Lack of clarity in the interpretation of the images in the draft's text 

2. The authors stated that fifteen patients (11 men and 4 women) were chosen for the study, but table 1 row 3 states sex (male) and the results clearly add up to fifteen in the same row. Is this just data from male patients? 

3. The total number of male and female patients selected clearly differs. How do the authors justify the gender-based results when the proportion is not equal or close to equal?

Author Response

Response to reviewer’s comments

Reviewer 3:

  1. Lack of clarity in the interpretation of the images in the draft's text

Response) Firstly, the authors appreciate the reviewer 3 for reviewing our manuscript. The text of Figures 2 to 4 was revised and the position of Figures and Tables were adjusted for easily understand the manuscript. Thank you for the reviewer’s understanding.

  1. The authors stated that fifteen patients (11 men and 4 women) were chosen for the study, but table 1 row 3 states sex (male) and the results clearly add up to fifteen in the same row. Is this just data from male patients?

Response) I am sorry for the confusion. Both men and women number were added to Table 1.

  1. The total number of male and female patients selected clearly differs. How do the authors justify the gender-based results when the proportion is not equal or close to equal?

Response) Thank you for the reviewer’s important comment. As the reviewer pointed out, there is clearly gender imbalance in this study. This is one of the limitations of this study. Therefore, the following description was added to the Limitation section.

Lines 410 to 412:

Thirdly, there was an imbalance in the number of male and female patients in this study. Therefore, the possibility that gender differences affect the efficacy of PRPr treatment cannot be denied.

Round 2

Reviewer 1 Report

The article is a very valuable one, so I would like to kindly request that in future articles the authors do not introduce elements that may affect the originality of the study.